# Advances in Diagnostic Approaches and Therapeutic Management in Bovine Mastitis

**DOI:** 10.3390/vetsci10070449

**Published:** 2023-07-08

**Authors:** Savleen Kour, Neelesh Sharma, Balaji N., Pavan Kumar, Jasvinder Singh Soodan, Marcos Veiga dos Santos, Young-Ok Son

**Affiliations:** 1Division of Veterinary Medicine, Faculty of Veterinary Sciences & Animal Husbandry, Sher-e-Kashmir University of Agricultural Sciences & Technology of Jammu, R.S. Pura, Jammu 181102, India; savleenkour.j13bv833@gmail.com (S.K.); balang248@gmail.com (B.N.); 2Department of Livestock Products Technology, College of Veterinary Science, Guru Angad Dev Veterinary and Animal Sciences University, Ludhiana, Punjab 141004, India; pavankumar@gadvasu.in; 3Division of Veterinary Clinical Complex, Faculty of Veterinary Sciences & Animal Husbandry, Sher-e-Kashmir University of Agricultural Sciences & Technology of Jammu, R.S. Pura, Jammu 181102, India; soodanjasvinder@gmail.com; 4Department of Animal Sciences, School of Veterinary Medicine and Animal Sciences, University of São Paulo, Pirassununga 13635-900, São Paulo, Brazil; 5Department of Animal Biotechnology, Faculty of Biotechnology, College of Applied Life Sciences and Interdisciplinary Graduate Program in Advanced Convergence Technology and Science, Jeju National University, Jeju 690756, Republic of Korea

**Keywords:** mastitis, advanced diagnosis, management, ethnoveterinary

## Abstract

**Simple Summary:**

Mastitis refers to the inflammation of the mammary parenchyma, caused by more than 136 microorganisms. It poses several challenges for farmers, veterinary clinicians, and researchers to understand and determine the most effective diagnostic tools and treatment protocols. This article discusses the clinical relevance, the causative pathogens, the economic factors involved, basic and advanced diagnostic techniques, and alternative therapeutic protocols required to control mastitis economically. There is a need to apply novel therapeutic technologies to overcome the challenges of traditional antibiotic-based therapies. These alternative therapeutic options could be supportive or additional options alongside conventional antibiotics-based therapies.

**Abstract:**

Mastitis causes huge economic losses to dairy farmers worldwide, which largely negatively affects the quality and quantity of milk. Mastitis decreases overall milk production, degrades milk quality, increases milk losses because of milk being discarded, and increases overall production costs due to higher treatment and labour costs and premature culling. This review article discusses mastitis with respect to its clinical epidemiology, the pathogens involved, economic losses, and basic and advanced diagnostic tools that have been used in recent times to diagnose mastitis effectively. There is an increasing focus on the application of novel therapeutic approaches as an alternative to conventional antibiotic therapy because of the decreasing effectiveness of antibiotics, emergence of antibiotic-resistant bacteria, issue of antibiotic residues in the food chain, food safety issues, and environmental impacts. This article also discussed nanoparticles’/chitosan’s roles in antibiotic-resistant strains and ethno-veterinary practices for mastitis treatment in dairy cattle.

## 1. Introduction

Mastitis refers to the inflammation of the mammary glands, involving changes in the gland tissue and glandular secretions causing physical and chemical alterations, respectively. The invasion of pathogenic organisms via the teat canal could be associated with contamination from the environment, unhygienic conditions, and, rarely, from systemic infections [1,2,3,4,5].

Mastitis can be classified into different categories viz, duration, symptoms, and pathogenic agent. In the context of the symptoms observed, the classification is broadly divided into clinical and sub-clinical forms. The clinical form of mastitis is characterized by rapid onset with swelling and redness of the affected quarter. The affected quarter may present physical and chemical alterations in the milk, containing flakes or clots, or having a watery consistency. In severe clinical mastitis, cows may present systemic signs ranging from visible lethargy to complete anorexia and high fever. In contrast, sub-clinical mastitis (SCM) largely goes undiagnosed due to difficulties in its diagnosis owing to the lack of visible changes in the milk, but the level of somatic cells exceeds 200,000 cells/mL [6]. This results in a marked loss in overall milk production [6].

The most prevalent causes of SCM are non-aureus Staphylococci, *Staphylococcus aureus* (*S. aureus*) and *Streptococcus* spp. The improper management of dairy cows has also contributed to the prevalence of SCM, such as high stocking density; stall feeding, poor hygiene and sanitation, such as cracked floors, poor drainage systems and dung piling up; flies; earth floors; and peri-parturient diseases [1]. As reported by Seegers et al. [7], SCM mastitis is 15 to 40 times more prevalent as compared with the clinical form, with a longer duration. Subclinical mastitis, therefore, serves as a carrier of pathogens that spread to healthy udders within the herd and is more challenging to prevent. As reported by Batavani et al. [8] and Bruckmaier et al. [9], an increase in the positive response in the California mastitis test (CMT) is accompanied by increases in sodium, chloride, IGF-1, and immunoglobulin, and decreases in calcium and inorganic phosphorus, showing the presence of tissue damage due to sub-clinical mastitis. Elevated levels of plasminogen, the activity of n-acetyl-β-D- glucosaminidase (NAGase), whey proteins, and γ-casein in the total protein has been reported in subclinical mastitis by Urech et al. [10].

Most mastitis cases are of bacterial origin worldwide, being caused by species such as S. aureus, *Streptococcus dysgalactiae* (*Strep. dysgalactiae*), *Streptococcus agalactiae* (*Strep. agalactiae*), *Streptococcus uberis* (*Strep. uberis*), and *E. coli* [11]. The microorganisms causing mastitis, such as *S. aureus*, *Strep. Agalactiae*, and *Mycoplasma* spp., spread from mastitis udders to non-mastitis (healthy) cows mostly through milkers’ hands and milking equipment. Machine milking acts as a reservoir and is a source of fomites carrying pathogenic bacteria. Algae of the genus Prototheca are usually among the causative agents of environmental mastitis, but, as per the report documented by Jánosi et al. [12] and Osumi et al. [13], it is not clear whether they are contagious or environmental pathogens. The major concern of environmental mastitis is faecal contamination and constant exposure of the teat canal, which remains open for 1–2 h post-milking. Meanwhile, contagious pathogens invade mammary glands during the milking process or by the colonization of teat skin [14].

The daily losses caused by mastitis in the first 2 weeks of lactation range from 1–2.5 Kg of milk, resulting in an estimated loss of 110–552 Kg. It has an ever-lasting effect on the milk yield as it is not possible to achieve peak milk production throughout their lactation period [15]. Despite several improved management practices in dairying, it is still a daunting disease condition causing huge economic losses to farmers across the globe. India ranked first among the milk-producing countries; hence, mastitis causes huge economic losses of about INR 575 million (estimated US$6.94 million) annually and a reduction in milk production by 21% [5]. Apart from farmer’s losses, the human health risks due to increased antimicrobial resistance and antibiotic residues in milk and milk products have decreased the demand for milk from the dairy sector. The consumer preference for natural/organic products has increased, as they believe that food produced from conventional farming systems is safer and healthier for consumption [16].

The association of mastitis and its causative pathogens was proven in the 1880s, with predominant pathogens identified during the 1950s. The multifactorial aetiology of bovine mastitis was found in the late 1960s, which paved the way for further research interest in this field [17], including common Gram-positive and Gram-negative microorganisms, such as *Strep. agalactiae*, *S. aureus*, *E. coli*, and *Klebsiella pneumoniae* (*K. pneumoniae*). The epidemiology and molecular characterization of pathogens at the subspecies level, virulence gene assays, whole-genome sequencing, and the in vitro susceptibility pattern of antibiotics were determined in the 20th century [18]. With the advancement of time, penicillin was made available for treatment by 1945, but it did not effectively work against all the pathogenic bacteria responsible for bovine mastitis.

Strategic managemental practices should be targeted during the dry period in order to minimize the incidence of mastitis after calving, particularly during the peak lactation period [18]. In heifers, subclinical mastitis is predominantly caused by minor pathogens during the peri-partum period, viz. coagulase-negative *Staphylococci* leading to mastitis. During early lactation, intra-mammary infections are influenced by many factors, viz. the onset time of calving, systemic disease, virulence gene array of the pathogen, cure, host immunity mechanism, management routines, herd location, season, parity, and peak milk production.

Definitive diagnosis of mastitis is very crucial for the dairy industry to ensure clean milk production, economic returns, public health concerns, and animal welfare compliance. Diagnoses should ideally be early, instantaneous, and rapid. They should specifically be used for management practices and early therapeutic interventions for mastitis. Over the last decade, the conventional method, viz. CMT, somatic cell count, the white side test, etc., have been in routine application for diagnoses of mastitis at the farm and individual cow levels. The conventional methods (e.g., CMT) are rapid, relatively cheap, and have field applicability, but have the disadvantage of non-specific detection. Advanced diagnostic tools, viz. polymerase chain reactions, protein-based ELISA techniques, acute phase protein detection, quantitative PCR, MALDI-TOF, etc., are costly, requiring skilled technicians, a laboratory, and sophisticated infrastructure. The great advantages of these tools are the highly accurate and specific nature of the detection of mastitis-causing pathogens, even at the subspecies level, providing an efficient method of treatment [19,20].

Along with management practices, numerous traditional and advanced novel therapeutic protocols are available for managing intramammary infection (IMI), including antibiotics, herbal therapy, bacteriocins, vaccination, and nanoparticle-based therapy [21]. Several compounds contribute to the prevention of mammary gland infections and also aid in increasing the lactation yield [22]. The most common treatment protocol used is antibiotic therapy. However, the uncontrolled and extensive use of antibiotic therapy and the persistence of biofilm-associated antibiotic resistance are responsible for poor antibiotic responses [23,24]. Vaccination programmes are one of the best methods for the prevention of specific mastitis at the herd level. The successful rate of vaccination against bovine mastitis is low because of the involvement of multi-etiological agents causing mastitis; nevertheless, *S. aureus*, *Streptococcus uberis* (*Strep. uberis*), and *E. coli* were considered as major targets for the development of vaccines [25,26,27]. There are a number of commercial vaccines on the market, but satisfactory results are still debatable [28]. As these therapies’ shortcomings have emerged, several advanced technologies have been introduced to fill the lacunae. Bacteriocins and nanoparticle-derived therapy are promising in providing protection [29,30]. This review manuscript critically analyses various aspects of intra-mammary infection/mastitis, focusing on its etiological agents, clinical importance, economic importance, advances in diagnosis, role of immunization, and therapeutic interventions.

## 2. Pathogen Causing Mastitis

Nocard and Mollereau, in 1887, appear to have been the first to undertake a study of the microorganisms causing mastitis [31]. They examined the udder secretions of ten cows suffering from severe contagious mastitis and successfully isolated streptococci (*Streptococcus mastitidis contagiosae*) in all cases. Pathogens causing bovine mastitis belong mostly to the environment and are ubiquitous in nature, whereas SCM is predominantly caused by contagious agents [32,33]. Some pathogens cause acute, per-acute, sub-acute, and chronic mastitis, categorized into either contagious or environmental forms of transmission. The most prevalent bacteria are *S. aureus*, *S. pyogenes*, *Strep. agalactiae*, *Enterobacter aerogenes*, *K. pneumoniae, Trueperella pyogenes* (*T. pyogenes*), *K. oxytoca*, *E. coli*, and *Pasteurella* spp. [34,35,36,37]. The major environmental pathogenic bacteria belong to the Enterobacteriaceae family, particularly *E. coli* [38], while contagious pathogens include *S. aureus*, *Strep. dysgalactiae*, and *S. agalactiae*. In clinical mastitis, *Strep. agalactiae* is the most frequent Gram-positive bacterium, followed by *S. aureus*, whereas *Klebsiella* spp. and *E. coli* are the most prevalent isolated Gram-negative bacteria in Brazil [39]. The main pathogens spread through direct contact are *S. agalactiae* and *S. aureus*; therefore, herd biosecurity is regarded as an important safety measure for the reduction and/or purging of their reservoirs [40]. The bacteria *T. pyogenes* is considered to be the sole cause of the clinical form of mastitis [41]. The highest level of loss caused to primiparous cows is caused by *S. aureus*, *Klebsiella* spp., and *E. coli*, whereas, in pluriparous cows, significant losses in milk production are caused by *Streptococcus* spp., *S. aureus*, *Klebsiella* spp., *T. pyogenes*, and *E. coli* [42]. In summary, the common pathogens causing mastitis are *S. aureus*, *Strep. agalactiae*, and *Strep. uberis*, whereas Mycoplasmal and *Corynebacterium* infections are less frequently diagnosed [43,44]. The heaviest forms of infection in intramammary tissues are associated with CAMP-negative *Streptococcus* spp., coliforms, *Strep. agalactiae*, fungi (yeast), *T. pyogenes*, and *Prototheca* spp. [41,45]. In a study of sub-clinical mastitis by Steele and McDougall [46], *Corynebacterium* spp. (40%) and *S. aureus* (32%) were the most prevalent bacteria in New Zealand. Opportunistic pathogens, such as the algae *Prototheca* spp., cause mastitis in lactating cows and also possess zoonotic capability [47,48]. Flies are of great importance in relation to animal and public health concerns because they act as mechanical vectors of many kinds of pathogens, such as bacteria, protozoa, viruses, and helminth eggs. Biting flies, including stable flies and horn flies, cause direct damage to dairy animals from direct blood loss, tissue damage, and allergic reactions [49]. A high level of activity of these flies reduces milk production in dairy animals [50,51,52]. In addition to their painful bites, horn flies transmit the pathogenic bacteria *Staphylococcus aureus* [53]. The possible interrelationships of genotypes of *Staphylococcus aureus* found in mammary glands and horn flies were documented by Anderson et al. [54], depicting the role of flies as an important source of bacteria. *Staphylococcus epidermidis* has also been isolated from fly traps by Woudstra et al. [55], but they did not detect the same strain in the milk and in the flies. Species of pathogens that have been detected in houseflies from dairies include *Enterococcus faecalis*, *hirae*, and *faecium*; *E. coli* [56]; and *Klebsiella pneumoniae* [57]. The identification of pathogen-causing mastitis has been considered as one of the steps forward for prevention and treatment purposes. With identified pathogens, specific antibiotic treatments can be used to avoid resistance due to their misuse. Many documented research articles from past decades to recent times have provided information about the potential causative agents of mastitis in livestock.

Common etiological agents causing mastitis in dairy cattle are documented in Table 1.

## 3. Clinical Relevance of Bovine Mastitis

Epidemiologically, mastitis is categorized as environmental and contagious forms, which are caused by a wide etiological pathogen [32]. Environmental factors, such as increased humidity with organic matter in the barn/shed area, significantly increase the bacterial load in the herd population. In one study, at the herd level, 74.7% prevalence, and at the individual level, 62.6% prevalence of mastitis was reported. In relation to the form of mastitis viz. sub-clinical and clinical, the former type has the highest prevalence of 59.2%, and the latter has a prevalence of 3.4% [32]. The clinical signs associated with evident symptoms include udder redness, increased size, pain to the touch, milk clots, discolouration, and abnormal milk consistency, along with systemic signs (pyrexia (>39.5 °C) and loss of appetite). In the Netherlands, among 20,000 clinical mastitis cases, *S. uberis* and *S. dysagalacticae* cause 40% of infections, and *S. aureus* and *E. coli* each cause 30% [142].

The economic aspects and incidence of clinical mastitis were studied by Kumar et al. [143]. In contrast to the clinical form of mastitis, no abnormal milk was observed in sub-clinical mastitis, although the change in the chemical composition of milk was an indicator for its diagnosis. It is diagnosed by laboratory tests of milk (composition) and animal-side tests, such as the California mastitis test (CMT), followed by the use of microbiological cultures for the identification of the causal pathogenic agent. Apart from CMT and microbiological culture isolation, somatic cells are considered a more reliable test for detecting sub-clinical mastitis, mostly white blood cells viz, infiltrated neutrophils, and macrophages in affected mammary gland tissues during inflammation in mastitis [144]. A study reported persistently higher levels of SCC in *S. agalactiae* infection, which was localized mainly in the udder [40].

Disturbance to the host immune response to infectious agents affecting intra-mammary glands is a major factor of mastitis [145]. Interferon-gamma and TNF-alpha are crucial components of innate and adaptive immunity against infectious pathogens and are important macrophage activators [146]. Additionally, the enhanced expression of these cytokines can induce a pro-inflammatory environment and facilitate oxidative damage, including increased free radical production. The relative number of T-cells, natural killer cells, and monocytes in peripheral blood increases during the post-calving period.

In a healthy udder, a balance of microbiota is important for maintaining the integrity of mammary gland homeostasis. The microbiota of the intra-mammary tissue is composed of a diverse community [147,148]. However, disrupting the diverse udder microbiota impacts the host’s immune response towards infection. Additionally, the normal microbiome of the udder is important to consider in making diagnoses of mastitis, as a healthy quarter also contains some bacteria. The possible bacterial genera present in udder microbiota include *Oscillospira*, *Ruminococcus*, *Roseburia*, *Dorea*, *Preotella*, *Baterioides, Bifidobacterium*, etc. Even any congenital or acquired anomalies viz. teat spiders, fistulas, and udder wounds that cause milk retention and bring forth the udder tissue to the external source of bacteria tend to cause mastitis [149]. In a histopathological study of mastitis-affected mammary tissue, an increase in the stromal connective tissue along with neutrophilia and a significant decrease in the alveolar epithelium was observed [60]. In clinical mastitis, the microbiota is mainly composed of *Staphylococcus* spp. and *Enterobactericaeae* family spp., disrupting the normal microbiota. Many researchers proposed that either the prolonged use of antibiotic therapy or alteration of the normal microbiome by the pathogenic agent are responsible for the establishment of mastitis [150]. In a study conducted in the USA, a total of ten negative coagulase *Staphylococcus* species (CNS) were isolated at different stages and seasons of the lactation period of dairy cattle [151].

Intra-mammary infection is a complex, yet detrimental, condition resulting from various host and environmental factors at the individual cow level. These involve pathogenic growth in mammary tissue, the host immune response (local and systemic), signalling pathways of various pro and anti-inflammatory cytokines establishing clinical outcomes, and various pathogen-associated molecular pathways. A possible approach to this is the recognition of toll-like receptors (TLRs), pattern-recognition patterns (PRPs), RIG-like receptors (RIGs), and NOD-like receptors (NLRs) in the evaluation of udder inflammation, either because of microbial or environmental causes. Hence, collaborative approaches for advanced diagnoses and for prophylactic measures of this disease are important [152].

## 4. Economic Significance

Mastitis leads to huge economic losses for dairy farmers, especially by affecting the overall milk quality and quantity. It also causes economic losses because of discarded milk, culling of affected animals, additional treatment costs, and extra labour costs. The projected annual economic loss caused by mastitis (subclinical and clinical mastitis) in India was reported to be US$ 98,228 million (INR 71,655.1 million) [6]. Significant overlooking of 60 to 70% of the total losses incurred due to SCM, which causes three times more production loss as compared with clinical mastitis, was reported [4,153]. Sinha et al. [4] reported higher losses in cross-bred cattle due to higher production levels, about 49% due to the value of milk and 37% due to veterinary expenses, followed by the cost of treatment (31%) and services (5.5%). Jingar et al. [154] reported a greater loss in indigenous Sahiwal and Tharparker cows (INR 1695.00), followed by crossbred (INR 1597.64) and Murrah buffaloes (INR 1498.44). In a meta-analysis, the prevalence of subclinical and clinical mastitis at the cow level was 41% and 27%, respectively, in India, indicating the importance of SCM [155]. In 2017, Rathod et al. [156] estimated that the costs for the loss of subclinical mastitis in India were about INR 21,677 to INR 88,340 per animal for a lactation period.

Many reports have suggested that India’s north and south zones show greater prevalence of SCM and CM with *Staphylococcus* spp. (45%) when compared with *Streptococcus* spp. (13%) and *E. coli* (14%), as the cattle population is greater in these zones. The overall prevalence of SCM in an updated meta-analysis by Krishnamoorthy et al. [157] was found to be 45%. Considering the zonal prevalence, Madhya Pradesh and Chhattisgarh (central zone) showed higher prevalence of subclinical mastitis (63% and 48%, respectively). Economic analysis of mastitis was conducted for 59 dairy farmers and an average drop in the milk yield during mastitis from 9–10 Kg to 6–7 Kg was found [158]. This caused an average loss in income from INR 413–458 to INR 306–335 per cow per day.

An economic evaluation of mastitis control under different intervention scenarios, quantifying the total cost of mastitis caused by *S. aureus* in Holstein cows in Argentina, was performed by Richardet et al. [159]. A total of 97.5 cases of *S. aureus*-caused mastitis in every 100 cows per year was estimated, and losses due to CM and SM were US$221.0 and US$151.7 kg/case, respectively. The higher probability of transmission increased the total cost of mastitis, caused by the large number of culled cows. This led to a drop in the economic efficiency of control and prevention programmes. Conversely, a decreased transmission rate apparently decreases milk losses due to mastitis.

Several studies on the prevalence and economic impact of mastitis indicate a need for comprehensive economic assessment as being of the utmost importance for formulating the various livestock health intervention efforts at present. A retrospective study using available animal health and dairy herd improvement records was conducted by Puerto et al. [160]. The authors used data from the first lactation of Holstein cows from 120 herds between 2003 and 2014 and assessed the production performance (in terms of overall milk, fat, and protein yield) and economic performance (in terms of milk economic value, margin over feed cost, and gross profit). The authors noted a significant decrease in the total milk yield (−382 to −989 kg) in mastitic cows.

## 5. Advances in Diagnostic Approaches

The diagnosis of clinical mastitis on a field basis is usually based on the detection of abnormal milk secretions, on the appearance of teats and udder symmetry, and palpation of the udder, indicating normal/abnormal consistency of the gland and signifying either an acute or chronic condition.

The early identification of subclinical mastitis is of paramount importance to prevent udder and milk loss; the present detection procedures, including Somatic cell count, electrical conductivity, and CMT, are less reliable as they depend on various stress-related pathways other than the causes of infection. An increased somatic cell count, positive California mastitis test, increased levels of enzymes (viz. N-acetyl beta-D-flucosaminidase; NAGase), lactate dehydrogenase; LDH) [46], along with the electrical conductivity of milk indicate a sub-clinical mastitis condition. The California mastitis test (CMT) provides a rapid and sensitive cow-side screening test to predict subclinical mastitis (SCC >200,000 cells/mL). The sensitivity and specificity of the CMT with the major mastitis pathogens (*Staphylococcus aureus*, *Streptococcus* spp., and Gram-negative organisms) during early lactation were 82.4% and 80.6% on day 4 of lactation, respectively [161].

Bacterial culture techniques are the gold-standard method to identify mastitis-causing pathogens, but need standardized, repeatable methods for their widespread application. Most of the pathogens readily grow on agar medium under aerobic conditions, but some microorganisms, such as *Mycoplasma* spp., need specific growth media. Culture techniques have limited sensitivity, which is further restricted as they require the isolation of one colony-forming unit (CFU) of the pathogen from 0.01 mL of milk (100 CFU/mL). The prevailing recommendation for considering a single quarter sample positive for an IMI is to use 100 CFU/mL, whereas that for non-aureus *Staphylococcus* is 200 CFU/mL [162]. Cultural isolation was considered as the gold standard for intramammary infection diagnosis. However, the increased specificity and sensitivity of PCR-based techniques have made it the new gold standard for the diagnosis of mastitis [163]. There are various tests used for the early detection of mastitis under field conditions and in the laboratory, as mentioned below (Figure 1).

### 5.1. Milk Somatic Cell Count (SCC)

The somatic cell count (SCC) provides an in-depth view of the quality of milk. The measurement of milk somatic cells by direct microscopy or using an automatic electronic cell counter still remains the most prevalent and easy method for the diagnosis of SCM. Conventional direct microscopy is labour-intensive, slow, and needs a high-quality microscope and trained personnel for improving efficiency. An automatic electron cell counter measures cell counts based on the principle of flow cytometry. It proves a rapid, sensitive, and accurate method for measuring the SCC [6,164]. Further, to reduce the labour and cost of diagnosis, composite samples for SCC at the cow-level are recommended for isolation from healthy cows. The sensitivity and specificity of the composite milk SCC as an indicator of mastitis in at least one-quarter range from 30 to 89% and 60 to 90%, respectively [165]. The most accurate correlation between IMI and SCC is found when analysing it at the quarter level. Further, data suggest that healthy quarters have a mean SCC of approximately 70,000 cells/mL, and an SCC of > 200,000 cells/mL indicates infected quarters [166,167].

### 5.2. Polymerase Chain Reaction (PCR)

Molecular markers are used to identify infectious pathogens that are difficult to isolate, i.e., *Mycoplasma* spp. [168]. Various DNA/RNA extraction methods for subsequent amplification with specifically designed primers are used for the molecular characterization of mastitis-causing pathogens [169]. Shome et al. [170] developed and evaluated a multiplex PCR for bacterial species (*n* = 10) causing mastitis in cattle, namely *S. aureus*, *Staph. chromogenes*, *S. epidermidis*, *S. sciuri*, *S. haemolyticus*, *S. simulans*, *Strept. agalactiae*, *Strept. dysgalactiae*, *Strept. uberis,* and *E. coli*. However, real-time PCR can provide results at a faster rate with greater sensitivity and specificity. A study by Ding et al. [171] developed a multiplex RT-PCR assay to detect *Staph*. *aureus*, *Listeria monocytogenes*, and *Salmonella* spp. in raw milk. The detection limit for the pure culture was 102 CFU/mL. Multiplex RT-PCR assay kits are commercially available on the market to detect *Strept. agalactiae*, *Strept. uberis*, and *S. aureus* with an accuracy of 98% [172]. Pathoproof, Thermo Fischer Scientific, Ltd. Waltham, MA USA commercialized the PCR kit for mastitis. Other kits, including DNA Diagnostic (Risskov, Denmark) and Mastitis 4, have also been made available. These kits work on the principle of RT-PCR with the quantification of the DNA of bacteria. The assay is highly accurate (95%), with sensitivity and specificity of 100% and 99–100%, respectively, at the udder quarter and cow levels [173,174].

Nevertheless, there is the potential to obtain false-positive results when utilizing PCR methods to identify pathogens in samples of mastitic milk. This could be because milk from healthy and mastitic cows is mixed. Furthermore, PCR cannot distinguish between viable and non-viable bacteria. Owing to all of these factors, it is advised that dairy advisors should use all related information at their disposal, including the history of the mastitis, udder inspection (clinically), previous treatment protocol, and SCC, together with the PCR results, to make the correct decision [173].

### 5.3. Nanotechnology and Biosensor-Based Diagnosis

Recently, nanotechnology and biosensor-based diagnosis methods have increasingly been used in the diagnosis of mastitis. They could be potential methods for the rapid and accurate diagnosis of various pathogens causing mastitis. Nanotechnology-based biosensors introduced the idea of having a laboratory “on a chip” [175]. Mujawar et al. [176] developed an advanced nanoparticle-based assay using three-dimensional nitrocellulose with microarray diagnostics. This assay uses black carbon nanoparticles and protein to prepare a conjugate for the secondary signal when tagging specific antibodies immobilized on the membrane. Flatbed scanning is used to detect mastitis pathogens in less than three hours (*S. aureus*, *Corynebacterium bovis*, *M. bovis*, *Strept. agalactiae*, *Strept. dysgalactiae*, and *Strept. uberis*). The assessment of protease activity by the colourimetric assay was utilized in identifying mastitic milk by Chinnapan et al. [177]. Magnetic nanoparticles along with the attached plasmin substrate form a black self-assembled monolayer on a gold sensor surface, and, on the cleavage of the substrate, peptide fragments attached to magnetic beads are released. The peptide fragments are attracted to the magnet in the sensor strips, leading them to the golden surface in the presence of an increased plasmin level. The sensitivity of this method is assessed at 1 ng/mL of plasmin in vitro. Similarly, aptamer–oligonucleotide or short peptides have been used to detect catalase in mastitic milk [178] with a sensitivity as low as 20.5nM [179]. Ribosomal protein (RP)-L7/L12 belongs to the 50S ribosome, expressed in microbes containing specific sequences for individual species. Their level increases in proportion to bacterial growth. It is highly specific for bacteria and can be used for rapid diagnoses as a target agent. In an experiment by Nagasawa et al. [180], these targets were used to develop anti-RP-L7/L12-coated immuno-chromatographic tests (ICS; colloidal gold nanoparticle-based immunochromatographic strips). The ICS reacted to *S. aureus* in a bacteria load-dependent procedure with 104 CFU/mL. Positive correlations have been observed between *S. aureus* (nuc gene) copy and test scores of ICS in mastitis milk.

### 5.4. Enzyme-Linked ImmunoSorbent Assay (ELISA)

The ELISA was first developed as a modification of RIA, and two research teams invented a direct form of ELISA simultaneously, named Eva Engvall and Peter Perlman, and Van Weemen and Schuurs. They devised it by tagging antigen and antibody radioisotopes in RIA with the help of enzymes, rather than radioactive iodine [125]. This method was employed to determine the level of IgG in the serum of rabbits [181]. The application of indirect ELISA to detect *Mycoplasma* infection in milk samples collected from herds has been performed previously [182,183]. Several studies evaluated this domain to develop an advanced tool for the earliest diagnosis, viz. a biomarker-based Liquid Phase-Blocking ELISA for subclinical mastitis [184], an indirect ELISA for detecting the antibody against *Streptococcus agalactiae* rAP1-BP-AP2 proteins and rSip-PGK-FbsA fusion protein [185,186]. Markers indicating the inflammatory response during udder infection have been assayed with the ELISA for the last decade. The detection of cytokines, such as tumour necrosis factors and interleukins [89,187,188,189,190,191,192,193], and acute-phase proteins, such as haptoglobin [72,194,195,196], with the ELISA has been considered as an important marker for the identification of mastitis in bovines. Recent advances in the field of ELISA have been made for specific and more sensitive assay detection, i.e., digital ELISA [197], ELISpot (Cecil Czerkinsky’s group in Gothenburg, Sweden (1983), Plasmonic ELISA (Nano-ELISA), providing ultra-sensitive and efficient detection methods [198], and sphere coated/bead ELISA [199]. Aptamer-based ELISA, with numerous target proteins, has more than one binding site, which enables them to form complexes with more than one recognition molecule, producing sandwich-like complexes. Similarly, aptamers are utilized in these ALISA assays, analogous to conventional ELISA [200,201]. Aptamer-coated magnetic beads and antibiotic-capped gold nanoclusters have also been applied for mastitis detection by Cheng et al. [202]. A SOMAmer (slow rate-modified aptamer) made from ssDNA containing pyrimidine residues and long dissociation rates was identified by Baumstummler et al. [203] to detect *S. aureus*. The SpA and ClfASOMAmers can selectively identify *S. aureus*. The ELONA (enzyme-linked oligo-nucleotide assay) was used to detect the proteinA-binding aptamer PA#2/8in *S. aureus* [204]. *Pseudomonas aeruginosa* was detected by using a glassy carbon electrode developed by Roushani et al. [205]. An aptamer-based electrochemical probe can be used for the detection of *P. aeruginosa* with a lower detection limit of 33 CFU/mL in samples. The sandwich ELISA assay was designed to generate two different aptamers against one target entity. These molecules can be used several times without a decrease in sensitivity and minimal non-specific absorption onto the platform’s surfaces [206]. Wide diversification of aptamers has been developed to detect *S. aureus* cells/their toxins (SEA, SEC1, SEB, and α toxin), along with various proteins (e.g., teichoic acid, peptidoglycan, etc.).

FRET (fluorescence resonance energy transfer) biosensors have high sensitivity for toxins and can spot SEA protein at a concentration of 8.7 ng/mL in milk, whereas graphene oxide (GO)-based optical biosensors have the ability to detect *S. aureus* at a minimum level of 8 CFU/mL [207]. Multiple and portable ELISA is a novel technique used for identifying pathogens/toxins and oncological markers. It consists of a multi-catcher device with 8–12 immunosorbents protruding pins onto a central stick that can be immersed into the sample. These ready-to-use lab kits are cost-effective, sensitive, allow the screening of large numbers of samples, and have no requirement for trained personnel or sophisticated equipment [208].

### 5.5. Proteomic-Based Diagnosis

By applying proteomic-based methods, pathogens could be diagnosed rapidly, sensitively, and accurately. Mass spectrometry is widely applied for identifying molecules based on their mass: charge ratio. The ion source in MALDI-TOF mass spectrometry is matrix-assisted laser desorption/ionization and the time-of-flight (MALDI-TOF) is the mass analyser (Figure 2). The MALDI-TOF technique is a proteomic-based approach that is gaining ground for the identification of bacteria in various disease samples, including mastitic milk [209]. Barreiro et al. [210] used MALDI-TOF MS for the identification of 33 bacterial isolates from bovine mastitis milk samples, and the results were compared with those obtained by classical biochemical methods. Barreiro et al. [211] identified *S. aureus*, *E. coli*, *Strep. agalactiae*, *Strep. dysgalactiae*, and *Strep. uberis* from experimentally contaminated milk samples (*n* = 15) with the use of MALDI-TOF MS coupled with Biotyper 3.0 software. Nonnemann et al. [212] identified isolates from (*n* = 473) samples of sub-clinical/clinical mastitis by the application of a thin smear from pure cultures onto a target plate covered withα-cyano-4-hydroxy cinnamic acid (HCCA) and submitted for MALDI-TOF MS identification. The isolates (CM or SCM, *n* = 413) displayed a variety of different bacteria: 19.1% *S. aureus*, 23.7% *Streptococci*, 15.6% *E. coli*, 15.6% non-aureus *Staphylococci*, 3.4% *Klebsiella* spp.; 2.9% *Corynebacterium* spp.; and 2.4% *Bacillus* spp. Additionally, the remaining isolates constituted about 17%.

Fidelis et al. [213] evaluated MALDI-TOF MS for the identification of Prototheca; yeast-like microalgae causing mastitis in dairy cattle, which are also nonresponsive to intramammary or systemic treatment with conventional antimicrobial agents, forcing the segregation and early culling of animals. MALDI-TOF identified 22 of 27 *P. bovis* and 3 of 4 *P. blaschkeae* isolates with scores of >2.0. With an extended algae database, MALDI-TOF MS can contribute to the speciation of *Prototheca* from mastitis cases quickly and easily. Many studies [214,215,216,217,218,219] evaluated the use of MALDI-TOF MS as an alternative method for the large-scale identification of conventional/non-conventional/antimicrobial-resistant bacteria isolates from milk samples of dairy animals compared with classical microbiological routine protocols with greater specificity and sensitivity.

## 6. Alternative Therapeutic Approaches for Bovine Mastitis

### 6.1. Nanoparticle-Based Therapeutic Interventions for Bovine Mastitis

Bacteria, such as *Staphylococcus aureus* and *Pseudomonas* sp., have the ability to form biofilms, which makes them more resistant to anti-microbials as they are packed in an extracellular polysaccharide matrix, giving them good protection from immune responses and anti-microbials. This decreases the efficacy of anti-microbials, providing a smaller therapeutic window [220]. The complications of an infected udder include the types of bacteria present producing various kinds of toxins and enzymes, which lead to tissue damage and increase the access of microorganisms to udder parenchymal tissue, thereby facilitating the survival of microorganisms in the keratin layer of the teat canal. Some strains of bacteria have protein A, which binds with the Fc portion of antibodies; this makes them unrecognizable by the neutrophil. Approximately 50% of *Staphylococcus aureus* produce beta-lactamase and cause micro-abscess formation. Some major virulence factors are responsible for therapeutic failure in *Staphylococcus* causing mastitis in dairy livestock. These factors make the penetration of antibiotics into the fibrous membrane difficult and complicated [221]. With all these factors included, major therapeutic failures in bovine mastitis are related to methicillin-resistant *Staphylococcus aureus* (MRSA). The probable cause of the resistance is the penicillin-binding protein coded by a genetic element termed the methicillin-resistant gene (mecA), which encodes a penicillin-binding protein 2a that is responsible for resistance against β-lactam antibiotics by blocking the β -lactam binding site [222]. Currently, the antibiotics used against MRSA include Daptomycin [223], Clindamycin [224], Quinupristin-dalfopristin [225], Tigecycline, a new analogue of tetracyclines that has in vitro activity against MRSA isolates [226], and Linezolid [227]. Despite the many anti-microbials available on the market, MRSA is a potential cause of economic losses because of mastitis in the dairy sector in terms of treatment cost and short-term effects. The deprived effectiveness of commonly used antimicrobials calls for immediate improvements in drug design, discovery, and delivery systems.

Chitosan (Qo) is a molecule of natural polysaccharide origin derived from chitin and is essentially composed of (β)-1,4 D-glucosamine linked to N-Acetyl-D-glucosamine residues [228]. It displays unique properties of biocompatibility and biodegradability, demonstrating it to be cost-effective, and provides alternative applications in food safety and biomedicine [229]. It has antibacterial properties, particularly against biofilm-producing Gram-positive bacteria, e.g., *S. aureus* [230], and Gram-negative bacteria, such as *E. coli* and *S. typhimurium* [231]. Many authors have conducted studies on the effect of chitosan in terms of the minimum inhibitory concentration (MIC), minimum bactericidal concentration (MBC), and disk diffusion assays against potential pathogenic agents, particularly the bacteria *Pseudomonas* spp. [232] and *Staphylococcus* spp. [221,233].

Nano-based drug-delivery systems introduce alternative therapeutics by effectively binding the drug to its larger surface area and carrying it to the target site with a controlled delivery rate (Figure 3). Different types of nanoparticles are used in the therapy of mastitis-causing pathogens, namely nanogels, solid lipid nanoparticles, liposomes, polymeric nanoparticles, and metal nanoparticles. Liposomes are spherical vesicles with one amphiphilic lipid bilayer with an aqueous internal core resembling the cell membrane. This bilayer with additional components, such as cholesterol or polyethene glycol (PEG), can be amplified with the objective of progress stability and biological compatibility [234]. Authors have documented many antibiotics loaded in liposomes, including encapsulated ciprofloxacin in liposomes [235], vancomycin-loaded liposomes [236], ceftazidime liposomes [237], levofloxacin liposomes [238], chloramphenicol-loaded deoxycholic acid liposomes [239], piperacillin, and β-lactam into liposomes [240]. Liposomes potentiate a drug’s pharmacological action, decrease drug toxicity, and are safer for parenteral administration. However, their stability is diminished due to the shorter shelf-lives of lipid vesicles and they are complicated to manufacture [241].

Polymeric nanoparticles have been frequently utilized in therapeutics and research work with promising antibiotic-delivery platforms. They are prepared to include hydrophilic or hydrophobic drugs, and macromolecules, such as nucleic acids, proteins, and peptides [242]. A report on the inhibition of the intracellular infection of *S. aureus* by tetracycline-loaded chitosan nanoparticles was documented by Maya et al. [243]. Similarly, the development of chitosan-coated iron oxide compound nanoparticles and levofloxacin-loaded calcium phosphate PLGA nanoparticles against the development of biofilm biomass was reported by Shi et al. [244] and Bastari et al. [245] respectively.

Nanogels are novel, three-dimensional, cross-linked nanocarriers used to release drugs with various mechanisms, such as thermo-sensitive, pH-responsive, and photoisomerization at the target sites. The size of nanogels ranges from 20 to 200nm. Nanogels are apposite and can be administered either with hydrophilic or hydrophobic drugs; owing to their smaller size, the invasion capability is increased and they also have a prolonged serum half-life [246]. Several types of nanogels are used for treatment against mastitis; vancomycin-loaded mannose hydrogel, which has an anti-MRSA effect [247]; silver nanoparticles-loaded dextran lysozyme nanogel [248]; gentamycin sulfate-loaded chitosan nanogel [249]; PLGA nanoparticles-loaded RBCs hydrogel, which neutralizes the toxin of *S. aureus* [250]; and rosemary essential oils-loaded chitosan benzoic acid nanogel [251].

Metal nanoparticles, such as silver nanoparticles, are considered as antibiofilm and antibacterial, and are even used against sub-clinical mastitis [252,253]. Gaseous nitric oxide (NO) has been found to be effective as an antimicrobial agent against Gram-negative and -positive bacteria, including MRSA. Friedman et al. [254] synthesized NO-releasing nanoparticles using chitosan. Similarly, Cardozo et al. [255] evaluated the S-nitroso-MSA-alginate/chitosan particles against MBSA and reported that NO-releasing polymeric particles are an interesting approach to overcome bacterial resistance in bovine mastitis treatment. In recent times, with the advancement of research in the field of nanoparticles, those with unique physiochemical properties and functionalization have remarkably aided in overcoming the restrictions posed by antibiotics. The evaluation of ciprofloxacin-loaded cerium oxide/chitosan nanocomposite synthesized using the seed extract of *Amomum subulatum* (black cardamom/BC) by Zafar et al. [256] against MRSA showed enhanced antibacterial activity compared with that of the free drug alone.

Ciprofloxacin from a nanocarrier is better sustained under physiological conditions, which could be considered as an effective and safe therapeutic protocol for MRSA-induced mastitis. Similarly, Aguayo et al. [232] manufactured chitosan nanoparticles with tripolyphosphate using ionotropic gelation and confronted *Pseudomonas* spp. isolated from clinically affected bovine mastitis. They showed a great antibacterial effect in the minimum inhibitory concentration (MIC), minimum bactericidal concentration (MBC), disk-diffusion assays, and biofilm inhibition. Additionally, Yadav et al. [257] demonstrated that the potential of ciprofloxacin-encapsulated nanocarriers against clinical strains of *E. coli* and *S. aureus* is dose-dependent (zone inhibition ranges from 11.6 to 14.5 mm and 15 to 18 mm when loaded with doses of 0.5 mg to 2 mg/mL nanoparticle solution). For the treatment of *E.coli*-affected mammary glands, various loaded nanoparticles have been found to be effective, including ciprofloxacin-loaded nanoparticles, which showed an antibacterial effect in heifers, screened from organized and unorganized farms in the Jammu region, India [258]; silver nanoparticle-decorated quercetin [259]; non-agglomerated ZnONPs (zinc-oxide nanoparticles) [260], gentamicin-and chloramphenicol-coated zinc oxide nanoparticles [261]; gold nanoparticles (AuNPs) using plant extracts (*Artemisia herba-alba* and *Morus alba*), which showed an antibacterial effect against multiple-drug-resistant *E. coli* [37]; and a nanohybrid of ciprofloxacin (CIP)–Ag (silver)–TiO_2_ (titanium oxide)–chitosan with biocompatibility with more than 93.08% of bovine mammary gland epithelial cells [256]. With the enlightenment of various important reports regarding the use of nanoparticles/ chitosan/ nanocarriers for the therapeutic management of bovine mastitis, there is a lot of documentation from many authors with respect to the processing, effectiveness, and applications of chitosan/nanoparticles, including Asli et al. [262], Breser et al. [263], Felipe et al. [264], and Zhang et al. [265]. There is still a need to discover new, safe, and cost-effective nanoformulations that can be available from the laboratory to clinical/field use, and this requires lots of research, guidelines, and production at a larger scale.

### 6.2. Herbal Therapeutic Interventions for Bovine Mastitis

Conventional and alternative therapies, including homoeopathy, have a significant role in veterinary medicine. In large animals, indications for homoeopathic treatment include Downer cow syndrome and mastitis, and colic in equines [266]. Treatment of clinical mastitis is, perhaps, the most common therapy in field conditions, and its response can be complicated with multi-etiological causal agents, antibiotic residues and resistance, and unavailability of broad-spectrum antibiotics in field conditions.

The clinical trials in Indian field conditions by various researchers on the effect of galactagogue comprising *Asparagus racemosus* and a mixture of *Leptadenia reticulate*, *Asparagus racemosus*, *Foeniculum vulgarae*, *Glycerhiza* spp., *Cyperus rotundus*, *Lepidium sativum*, *Cuminum cyminum*, etc. have shown appreciable galactagogue efficacy [267,268]. A similar study was conducted in the United Kingdom by Wheeler and Wait [269] with the collective use of galactagogue and antibiotics in SCM cases, and it restored milk loss due to mastitis. It was shown to increase milk production and restore pre-treatment lactation. Cure rates of 68.5, 78.2, and 91.1% (including 41.1, 71.0, and 71.4% bacteriological cure) were reported for the intramammary infusion of the herbal preparations Injecta CI, An-Ru, and Shuang-Ding, respectively, which were better than the cure rates obtained for penicillin and gentamicin treatments [270].

Non-antibiotic approaches eliminate residue problems and antibiotic resistance, and have become a topic of interest for public health and for research. Herbal preparations and acupuncture have been attempted in several institutes in China [270]. Better results were observed when administered intramammary along with chlorhexidine once daily. During this period, initial work focused on the non-invasive and non-irritant properties of herbal preparation. Another study with Injecta CI administered via intramammary infusion was compared with a penicillin–streptomycin-treated group. The injected CI had a cure rate of 74.1%, which covered bacteriologically diseased quarter in 43.5% (37/85) [270].

Hu et al. [271] studied the influence of herbal preparations on blood and milk phagocytes. Twenty-six herbal preparations considered as antipyretics in Chinese materia medica were assessed in vitro to evaluate their impact on the phagocytic potential of neutrophils labelled with 3ZP-labelled *Staphylococcus* isolated from bovine blood and milk. The active preparation from *Radix bupleuri*, *Folium hibisci*, *Herba houttuyniae*, *Flos chtysanthemi*, *Caulis lonicerae*, *Radix stellariue*, *Herbasenecionisscandentis*, and *Floslonicerae* increased phagocytosis by over 35.0%, simulating milk neutrophil functions. Intramammary preparations derived from herbal ingredients, such as An-ru prepared from *Oleum Eucalyptz* and Sbuang-ding from *Herbataraxuci* and *Herbaviolae*, are effective in mastitis treatment [272], but cause irritation to the udder. Therefore, we suggest further cytotoxicity studies and optimizing the dose before developing commercial intramammary products.

During the late 20th century, emphasis was placed on the immune potent properties of herbal plants against bovine mastitis and a clinical trial by Chishti et al. [273] on 327 cows and 493 buffaloes for the effect of immune potent herbal agents (Lasoni herbal powder; Almuslim Herbal Research Corporation, Faisalabad, Pakistan) against sub-clinical mastitis was performed. The herbal powder was not effective in the treatment of sub-clinical mastitis.

Nisin has been used for its preservative properties in the food industry for decades. It is an antimicrobial protein produced by the *Lactococcus* subspecies lactis, with lower molecular weight. Recent advances in the production of pure nisin have re-evaluated its use for pharmaceutical and veterinary purposes [274]. A wide range of anti-microbial activity against Gram-positive bacteria has been shown by nisin, with little or no activity against Gram-negative ones [275]. Nisin exhibits good potential as a therapeutic agent in treating bovine mastitis and is strongly bactericidal towards mastitis pathogens. In one of the studies conducted on intramammary infection, challenge infection of the teat was performed with the pathogen and followed by three intra-mammary treatments at 12 h intervals. High curative rates were achieved, with 66% for *S. aureus*, 95% for *S. agalactiae,* and 100% for *S. uberis* [274]. However, no significant correlation was found between the cure rate and the number of days of infection. Additionally, a significant reduction in the somatic cell count was observed. Nisin is a peptide, rather than an antibiotic, having a non-toxic effect with no residues, which makes it safe for consumption by consumers. Oral care applications have actively been explored in the beagle dog model as a mouth rinse [276]. The extensive sensitivity of *Streptococcus* and *Staphylococcus* species to nisin offers great opportunities for topical infection and multiple drug-resistant systemic diseases (MRSA).

Ethno-veterinary medicine in livestock management and rearing has been in dispute for a long time. A substantial proportion of professionals in various fields have documented, valued, and studied the potential effectiveness of ethno-centrally derived plants for traditional animal health care and practices in native communities. These medicines have been used to treat livestock health disorders, primarily mastitis. In Bangladesh, a study attempted to identify farmers’ use of ethnoveterinary medicines in the management and rearing of livestock by Islam and Kashem [277]. Out of 32 EVMs (ethno-veterinary medicine), the galactogogue effect was observed in dairy cattle by feeding Katanate (*Amaranthus spinosus*) and Shiru (*Leersia hexandra*), and feeding Mashur (*Lens esculenta*) bran, oil of Tishi (*Linum usitatissimum*), gum of tamarind, and Malbogh banana together with straw. Another preparation named Mastilep (*Glycyrrhiza glabra*, 5 g; *Curcuma longa*, 2 g; *Cedrus deodara*, 10 g; *Paederia foetida*, 5 g; and sulfur, 10 g, in gel base) was studied for its efficacy as a supportive therapy in clinical mastitis [278]. Mastilep was applied topically to the udder and teats along with parenteral antibiotics in 10 cows. Eight of the ten cows (80%) recovered after the parenteral administration of antibiotics combined with intramammary infusion. The recovery rates for intramammary antibiotics and the application of Mastilep alone were 40% and 30%, respectively. A similar trial was conducted by Sharma and co-workers [279] with Mastilep (*Glycyrrhiza glabra*, *Curcuma longa*, *Eucalyptus globulus*, *Cedrus deodara*, *Paederia foetida*, and sulphur), and reported great results with controlled infection irrespective of aetiological agent, increased milk production, and no re-occurrence of clinical mastitis within 2 months. The efficacy of topical Mastilep in the case of subclinical mastitis in crossbred cows was determined by Deepa et al. [280], and a significant (*p* < 0.05) decrease in the somatic cell count during mid- and late lactation was observed. The effect of Mastilep topical gel was assessed in small dairy units of the Khanapara region of Assam and Meghalaya, and recovery rates of 88.3% and 88% in two other farms were found by Nath and Dutta [281].

A possible remedial effect of *Persicaria senegalense* on sub-clinical bovine mastitis was studied by Abaineh and Sintayehu [282]. A significant cure rate was observed in the experimental group (0.77 kg leaf powder of *Persicaria senegalense*) against bacterial isolates of *S. aureus*, *C. bovis*, *C. albicans*, and *P. aeruginosa*, and a negative control group. Similarly, a survey was performed in the southern Rajasthan region for ethnoveterinary herbal medicine during 1999–2001 by Takhar and Chaudhary [283]. *Vernonia anthelmentica* (Family; *Astaeraceae*) seeds and their decotion with jaggery were found to be effective against all udder disorders. *Cistanche tubulosa* (*Orobanchaceae*) whole-plant paste and *Tachyspermum ammi* (*Apiaceae*; seed) were used for treatment against bovine viral mammillitis (BVM).

An aqueous extract of *Ocimum sanctum* L. showed immunotherapeutic potential against SCM [284]. It showed a decreased total bacterial count with increased levels of neutrophils and lymphocytes and enhanced phagocytic activity. Additionally, the lysosomal contents of polymorphonuclear cells of milk in animals treated with the extract were enhanced significantly. This result substantiated the medicinal herb’s therapeutic effect and emphasized the potential of non-toxic substances for udder immunity. The leaves of Hedera Helix containing alkaloids, flavonoids, glucoside, and organic acids have been documented amongst 41 remedial plants to playa role in treating post-partum disorders in cattle in Sardinia, Italy [285]. However, a single case remedy of Hedera Helix was published in the same document. Additionally, the fluid extract of *Spirea ulmaria* and *Astragalus membranaceus* against bovine sub-clinical mastitis was evaluated by Gianciti et al. [286]. The evaluation was performed on the basis of the presence of bacteria, SCC, and milk production. The results were particularly effective against coagulase-negative *Staphylococci* and significantly reduced quarter infection (16.7% vs. 30.2% and 37.5%, respectively, in control and placebo groups). In the same year, an observation was made regarding the activity of the selection component of Chinese herbal plants on blood flow stasis for dairy cow mastitis [287]. Decoctions of red sage root (RSR, *Radix salvia miltiorrhizae*), giant knotweed rhizome (GKR, *Rhizoma polygoni cuspidati*), chuanxiong rhizome (CXR, *Rhizoma chuanxiong*), and safflower (SF, *Flos carthami*) were developed.

A study of the herbal plant *Azadirachta indica*’s response to mastitis at the cellular level (expression of cytokine and respiratory burst activity of milk neutrophil) was performed by De and Mukherjee [288]. The results showed a significant (*p* < 0.05) decrease in the total bacteria count (TBC), somatic cell count (SCC), and milk neutrophil, and increased levels of lymphocytes, hydrogen peroxidase, and superoxide ion. However, cytokines (IL-2 and IFN- γ) were expressed normally in treated and normal healthy cows. This suggests the potential of herbs in terms of anti-inflammation, anti-bacterial, and immuno-modulation. In the consecutive year, a study was performed on the therapeutic activity and immunomodulation of *Tinospora cordifolia* against bovine sub-clinical mastitis by Mukherjee and co-workers, and found similar results for the TBC, SCC, and level of IL-18 in udders to those obtained for *Azadirachta indica*. *Azadirachta indica* has more than 135 bioactive compounds: isoprenoids containing limonoids, protomeliacins, gedunin, azadirone, vilasinin, and C- secomeliacins, such assalanin, nimbin and azadirachtin. These isoprenoids tend to have anti-bacterial and anti-fungal activity [289]. Anti-proliferative xylooligosaccharides (4-O-methyl glucuronic acid substitution) from the extraction of xylans in *Azadirachta indica* were reported by Sharma et al. [290]. They activate the intrinsic path way of apoptosis, indicating that xylooigosaccharides could potentially be used as anti-proliferative compounds.

The medicinal herb *Houttuynia cordata* Thunb, used to derive houttuynin sodium bisulphate (HSB), α-hydroxyl-capryl-ethyl-sodium-sulphonate, is a product manufactured by reacting sodium bisulphate with houttuynin for treating bovine clinical mastitis. It was found to be effective against acute and sub-acute mastitis clinically and microbiologically. It has a mild inhibitory effect on streptococci and the withdrawal time is 12 h post-treatment [291].

### 6.3. Role of Bioactive Compounds Presents in Herbal Remedies against Mastitis

There are potential herb extracts with antibacterial activity and protective ability to inhibit LPS-induced cell death and inflammatory responses. The identification of the bioactive compounds and their role in therapeutic measures against mastitis will provide an alternative approach. With the help of advanced diagnostic aids and alternative therapeutic approaches, one can control mastitis and associated antibiotic resistance.

Panya et al. [292] demonstrated the bioactive compounds in *Clinacanthus nutans* (Lindau) using solvent fractionation, HPLC, and LC-MS/MS analysis. They revealed that glyceryl 1,3-disterate (C_39_H_76_O_5_), kaempferol 3-O-feruloyl-sophoroside 7-O-glucoside (C_43_H_48_O_24_), and hydroxypthioceranic acid (C_46_H_92_O_3_) had great potential against mastitis. Yu et al. [293] demonstrated the anti-inflammatory effect of glyceryl 1,3-disterate (C_39_H_76_O_5)_ in LPS-stimulated macrophages by downregulating the expression of cyclooxygenase-2 (COX-2), inducible nitric oxide synthase (iNOS), and the inflammatory cytokinesinterleukin-6 (IL6) and tumour necrosis factor alpha (TNF alpha). The potential increase in TNF-alpha ultimately activates caspase, a reactive oxygen species (ROS) product that ultimately causes cell apoptosis. Lopez-Lazaro et al. [294] demonstrated the effect of kaempferol on modulating the anti-inflammation response via inhibiting NF-kB activity, strongly diminishing ROS production in response to H_2_O_2_, and significantly increasing cell viability, suggesting the role of kaempferol in apoptosis inhibition.

Morale-Ubaldo et al. [295] isolated and characterized the antibacterial compounds from *Larrea tridentate* through HPLC techniques against multidrug-resistant bacteria associated with bovine mastitis. The results indicated that nor-3 demethoxy isoguaiacin can be used as an alternative treatment for mastitis. It showed activity against the cell membrane by repressing proteins of the ATP-binding cassette transport system, which causes bacterial death [296].

The medicinal herb *Houttuynia cordata* was found to be effective against bovine acute and sub-acute mastitis. Yang and Wang [297] new sodium houttuyfonate (*Houttuynia cordata)* could inhibit the proliferation and promote the apoptosis of human ovarian cancer A2780 cells. The mRNA expression of the apoptosis-related molecules Bcl2, Bax, VEGF, and NF-κBp65 was detected by real-time PCR. The expressions of VEGF, Bcl-2, and NF-κBp65 in the group treated with neo houttuynia sodium decreased, and the expression of Bax increased.

The anti-microbial effects of Aloe vera on bacteria associated with mastitis in dairy cattle have been assessed by many authors in recent times, viz. Forno bell et al. [298,299]. Their study showed that Aloe vera extract disrupted the cell membranes, causing the lysis of *Staphylococcus aureus*, *E. coli*, *Streptococcus uberis*, and *MRSA* due to the presence of anthraquinones, such as aloin and aloe emodin.

*Ocimum tenuiflorum* L. has an essential volatile oil comprising phenols, terpenes, and aldehydes that are mainly concentrated in the leaf. The compounds, including rosmarinic acid, luteolin, and apigenin, were assessed for their phytochemical constituents by liquid chromatography–electrospray ionization–tandem mass spectrometry (LC-ESI-MS/MS), and showed a significant reduction in the immune response in macrophages by inhibiting the expression of pro-inflammatory cytokines (IL-6, TNF-α, and IL-1β) induced by LPS. They also decreased the LPS-stimulated expression of iNOS and COX-2 in a concentration-dependent manner. They can be used as synergistic antibacterial compounds with standard drugs in mastitis treatment [300]. *Ocimum sanctum* has a variety of constituents, such as saponins, triterpenoids, flavonoids, and tannins. The methanol extract and aqueous solution of holy basil leaves cause a significant increase in the levels of IFN-γ andIL-4, and percentages of T-helper cells and NK-cells; therefore, they have immunotherapeutic potential in bovine sub-clinical mastitis. Reduced total bacterial counts and increased neutrophil and lymphocyte counts with enhanced phagocytic activity were observed in mastitic cattle administered an extract of *Ocimum sanctum* [301]. Mukherjee et al. [284] demonstrated the in vivo efficacy of *Ocimum sanctum*, which revealed an increase in the neutrophil, lymphocyte, and lysosomal enzyme contents of the milk polymorphonuclear cells, thus enhancing mammary immunity.

## 7. Role of Immunization and Its Constraints

Immunization and improved managemental practices have traditionally been used as the main preventive approaches against mastitis [16,302]. Commercially, mono (against one pathogen) and polyvalent (against more than one pathogen) vaccines are available against the pathogens causing mastitis, viz. ENVIRACORTM J-5 Vaccine (*E. coli* J5 mutant bacterin) produced by Zoetis, Parsippany, New Jersey, USA, J-VAC^®^; *Escherichia coli* Bacterin-Toxoid (*E.coli* mutant strain is the J-VAC) produced by Merial, Ingelheim am Rhein, Germany, ENDOVAC; and the dairy coliform vaccine (bacterin-toxoid formulated from a Re-17 mutant of *Salmonella typhimurium*) produced by Immvac Inc. Columbia, MO, USA Vaccines against *S. aureus* are also available, such as Lysigin, produced by Boehringer Ingelheim Animal Health, St Joseph, MO, USA and *Mycoplasma bovis*, e.g., *Mycomune* bacterin, produced by AgriLabs, Inc. St. Bronson, MI, USA. On the other hand, polyvalent vaccines include STARTVAC^®^ (inactivated vaccine contains a mixture of *E. coli* J5 and *S. aureus* (CP8) strains SP 140); Hipramastivac, which contains *S. aureus* (TC5 and TC8 strains) and *E. coli* (J5 strain), in addition to *S. agalactiae*, *S. uberis*, *S. dsygalactiae*, *S. pyogenes*, *P. aeruginosa*, and *A. pyogenes* bacterins; and MastaVac (*Staphylococcal enterotoxin* Type C mutant vaccine) [303]. Zhylkaidar et al. [304] determined the effectiveness of polyvalent vaccines against *Staphylococci*, *Streptococci*, *Escherichia*, *Klebsiella*, *Diplococci*, and *Protea*. Out of 600 immunized cows, 9 (1.5%) and 13 (2.3%) animals developed subclinical and clinical mastitis, respectively. The antibodies in the blood serum exceeded the initial indicator by 30 to 40 times. The milk yield in vaccinated cows was at the same level as their previous lactation. However, the efficacy of two commercial vaccines (Startvac^®^, developed by HIPRA Spain, and Mastivac^®^, Laboratorios Ovejero, Spain) was analysed by Tashakkori et al. [305] and they did not find any significant changes in the incidence of clinical mastitis, somatic cell count, and anti-oxidants. The insufficient response to the vaccine could be attributed to many factors viz. age, health status, the invading pathogen and its strain, and variations in immune responses among individuals because of environmental and genetic variations [28].

Nanotechnology has been used and investigated by Nagasawa et al. [180] in the form of a nasal mastitis vaccine based on a conjugated cCHP nanogel and inactivated *Staph. aureus* antigens, which could release protective levels of anti-*Staph. aureus* IgA. The DNA vaccine encapsulated in chitosan NPs (pPCFN-CpG-CS-NPs) *Trueperella pyogenes* (*Arcanobacterium pyogenes*) was investigated in a mouse model. The data showed that it could provide protection against challenge with mastitis-causing pathogens. However, no data are available on its use and protection against intramammary invasions [306]. Quiroga et al. [307] assessed the efficacy of a novel vaccine against mastitis using proteoliposomes obtained from *E. coli* in a murine model of coliform mastitis. They demonstrated that the proteoliposome vaccine was safe, immunogenic, and effective against an experimental model of *E. coli* mastitis, decreasing the bacterial count and tissue damage. This proteoliposome vaccine could be used as a potential new tool for the prevention of mastitis.

Zeng et al. [308] evaluated the immunogenicity of KLH-Ent conjugate vaccine (Keyhole Limpet Hemocyanin-Enterobactin) in Holstein dairy cows at drying off (0 days) and 21 and 42 days after drying off. It significantly induced serum and milk Ent-specific IgG and IgG2 antibodies post-vaccination at calving and during early lactation at 14 and 30 d in milk. Based on its safety by monitoring rectal body temperature and injection site reactions, the KLH-Ent conjugate vaccine is safe for dairy cows. Similar results were obtained in the study by Wang et al. [309], with significantly increased (up to 4096-fold) anti-Ent IgG antibody titres in experimental rabbit serum. Additionally, Enterobactin is utilized by the *Enterobacteriaceae* family; it is crucial to monitor the effect of Ent-specific antibodies on the microbiota community composition in vaccinated and unvaccinated cattle. As reported by Zeng et al. [308], no significant difference (*p*> 0.05) in the faecal *E. coli* counts in CFUs/gram of faeces and in microbial community structures and diversity was foundbythe16S rRNA gene sequencing of faecal microbiota between vaccinated and control groups. These observations encourage the prevention of clinical mastitis, especially during the first 30 days of lactation, caused by *Coliform* bacteria.

A brief summary of key features of vaccines against mastitis are presented in Table 2.

Some of the effective constraints in the defence mechanism of mammary glands have been identified, such as the inhibitory effects of fat and casein on phagocyte functions, and the excellent growth medium provided by milk for many bacterial pathogens [318]. Phagocytosis by neutrophils in mammary glands is arguably the major defence mechanism against pathogens, and vaccination can improve opsonophagocytosis [319]. There are arguments that mastitis does not induce the production of immune memory cells. Additionally, subclinical mastitis may not be able to induce an immune response in some cases because the intensity of the stimulus by antigen is too low [320]. It has been proposed that the antigenic effect of chronic *S. aureus* infection on the memory type of the immune response in bovine mammary glands is minimal. The persistence of *S. aureus* infection may result, in part, from the suboptimal stimulation or immunosuppression of the mammary immune system. Live vaccines tend to elicit a different immune response, mainly an IgG_2_ antibody response and an amplified influx of neutrophils into the mammary gland, whereas killed *S. aureus* administered with oily adjuvants stimulates the production of predominantly IgG_1_ antibodies and results in no change in neutrophil recruitment at the onset of an experimental infectious challenge [321]. It is clear that natural IMI itself does not elicit complete protection against reinfection, so for effective immunization, vaccines need to improve the natural immune response. Vaccination with recombinant proteins elicits neutralizing antibodies, whereas natural infection does not. This can be beneficial for definitive diagnosis between vaccinated and naturally infected dairy cattle [322].

## 8. Conclusions

There is an urgent need to implement effective control and managemental strategies for mastitis in dairy cattle to improve dairy economics by ensuring proper quality and increasing the quantity of milk. A multipronged and holistic approach involving all stakeholders, including the dairy farm workforce, pharmaceutical industry, veterinary services, rapid and sensitive diagnostic tests at an affordable price, and government incentives is required to control it. Various ethnoveterinary and advanced natural therapies, such as using plant extracts, herbs, chitosan, nanogels, etc., could be a panacea for conventional antibiotic therapy, which poses public health and food security risks, such as the emergence of antibiotic-resistant bacteria, antibiotic residues in the food chain, and environmental issues. With the concept of the One Health approach, alternative therapeutic protocols should be promoted and popularized to reduce or replace the use of antibiotics.

## Figures and Tables

**Figure 1 vetsci-10-00449-f001:**
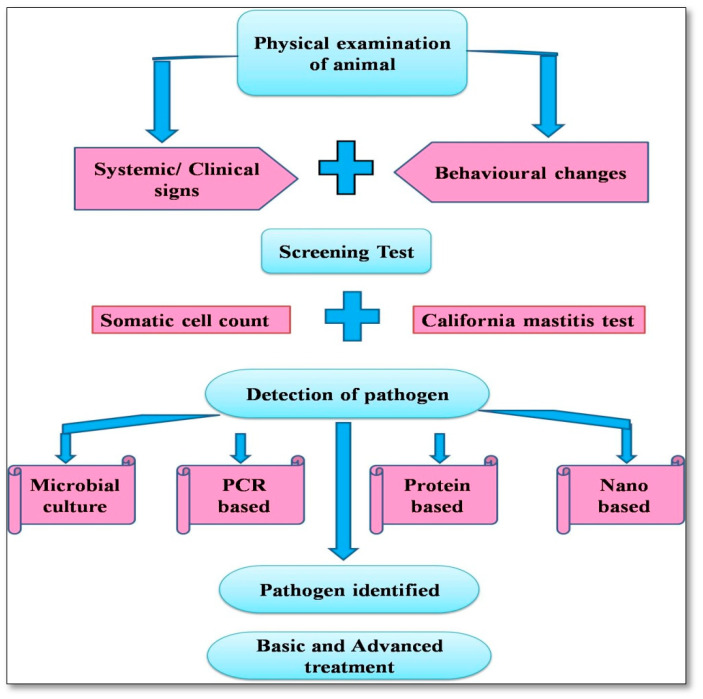
Overview of diagnoses of Bovine mastitis.

**Figure 2 vetsci-10-00449-f002:**
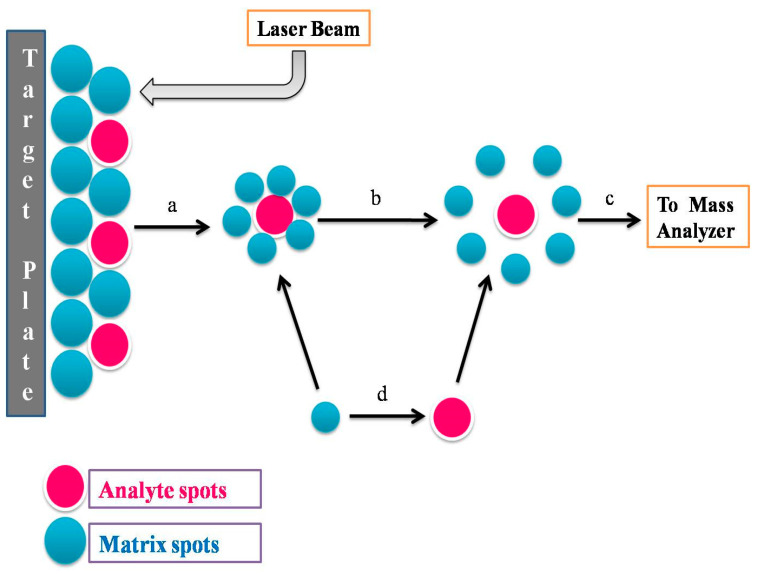
Ionization of analytes by MALDI; (a) desorption, (b) desolvation and ionization, (c) mass analyser, and (d) proton transfer.

**Figure 3 vetsci-10-00449-f003:**
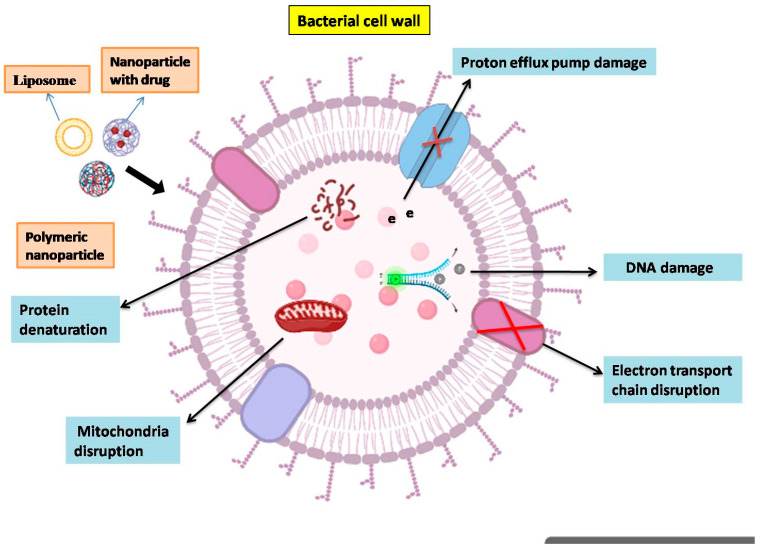
Cytotoxic effect of nanoparticles on bacterial cell wall; DNA damage, electron transport chain disruption, mitochondrial disruption, and proton efflux damage.

**Table 1 vetsci-10-00449-t001:** Common etiological agents causing mastitis in dairy cattle documented from past decades to recent times.

S No.	Causative Agent of Mastitis	References
1.	*Staphylococcus* spp.	
	*Staphylococcus* spp.	[45,58,59,60,61,62]
	*S. aureus*	[63,64,65,66,67,68,69,70,71,72,73,74,75,76,77,78,79,80,81,82]
	*S.mastitidis*	[58,83,84]
	*Coagulase-negative Staphylococcus* (CNS)	[65,85,86,87,88,89,90,91,92,93]
2.	*Streptococcus* spp.	
	*Streptococcus* spp.	[6,59,94,95,96]
	*Strep. agalacticae*	[11,97,98,99,100,101,102,103,104,105,106]
	*Strep. viridians*	[107]
	*Strep. lactis*	[108]
	*Strep. mastitidis, Strep. acidominimus*	[109,110,111]
	*Strep.uberis*	[97,100,112,113,114,115,116,117,118,119]
3.	*E. coli*	[6,63,66,67,74,96,118,120,121,122,123,124,125,126,127]
4.	*Bacillus* spp.	[67,69,126,128,129,130,131]
5.	*Tuberculous type bacilli*	[58,119,132]
6.	*Mycoplasma* spp.	[69,119]
7.	*Aerobacter aerogens*	[84]
8.	*Pseudomonas aeruginosa*	[69,97,119,133,134]
9.	*Corynebacterium* spp.	[135,136,137,138,139,140,141]

**Table 2 vetsci-10-00449-t002:** Brief summary of key features of vaccines against mastitis.

Key Features of Coliform Vaccine Trails			
Vaccine Antigen	Efficacy	Knowledge Gap	References
*Escherichia coliJ5* bacterins	Decreased coliform mastitis severity in field experiments	Variable effect on incidence of cases, unknown mechanism	[310]
*Salmonella* Re-17 bacterin toxoid	Decreased severity in field experiment	Unknown mechanism	[311]
*E. coli J5* bacterin with killed *Staphylococcus aureus* (StartVac, Hipra)	Less severity in field cases	Unknown mechanism	[27]
Whole *E. coli* (P4), intramammary booster with bacterial extract	Reduction in severity, likely independent of antibodies, related to Th17 response	Test for heterologous protection not performed	[191]
*Klebsiella* recombinant YidR	Reduced incidence of Kleibsiella mastitis	Unknown mechanism and little antibody response to whole bacteria	[312]
**Key Features of Staphylococcus Aureus Vaccine Trials**			
**Vaccine Antigen**	**Efficacy**	**Knowledge Gap**	**References**
Protein A (SpA)	Spontaneous cure of *Staph. Aureus* infection with experimental challenge	Mechanism not identified, no field trial	[313]
Killed vaccine, “in vivo” antigen and dextran sulfate	Severity of infection low	Mechanism not identified	[314]
Bacterial lysate (5 strains) Lysigin (Boehringer Ingelheim Vetmedica, Lyon, France)	Reduction in intramammary infection	Variable results	[315]
Recombinant IsdB and IsdH	IgG2 antibodies and antigen-specific lymphoproliferation	Protection study in cows not documented	[316]
Slime on killed bacteria, StartVac (Hipra)	Reduction in bacterial shedding in milk	Mechanism not identified	[317]

## Data Availability

The data presented in this paper are available on request from the corresponding author.

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
