# Peer review of "Advances in Diagnostic Approaches and Therapeutic Management in Bovine Mastitis"

_vetsci, 2023, doi:10.3390/vetsci10070449_

Round 1

Reviewer 1 Report

Comment to Authors

In general, the manuscript is well structured, data is presented and well discussed.

The paper is interesting, therefore are needed minor revisions.

Some points must be attended to before publication:

 Line 333-334 Please improve the sentence.

Line 388-391 Please improve the sentence.

Best regards

Author Response

Reviewer #1

Comments and Suggestions for Authors

Comment-1: Line 333-334 Please improve the sentence.

Response: Changes has been made as suggested

Comment-2: Line 388-391 Please improve the sentence.

Response: Changes has been made as suggested

Authors are grateful to the honourable reviewers for helping in improving the manuscript.

Reviewer 2 Report

I would like to thank the authors for the very good work done. However, I have some comments.

1. No mention had been done on the association between flies and summer mastitis on the therapeutic section of the manuscript.

2. Despite the plethora of published studies, there is a huge lack of information regarding the detection (through HLPC) of specific substances contained in herbs that may have these anti-mastitis effects and, of course, their mechanisms of action. For this reason, I recommend a different section or paragraph describing those important issues. I think it will differentiate your paper and will increase its validity.

3. Another issue that can complicate diagnostic and therapeutic approaches is vaccination (with commercial product or self-vaccination). It can interfere diagnosis through its producing antibodies (ELISA) and it can "hide" mastitic cases which demand special therapeutic approach. From my personal experience, vaccination made invisible some S. aureus cases and permit this microbe spread in all herd unnoticed before we take action against it. But the damage has done!

I kindly recommend the authors to address my comments and I would like to thank them in advance.

Author Response

Article id- Vetsci-2439808

First of all I would like to thank to all revered reviewers and editor for their critical insight to our manuscript. Reviewer’s suggestions have helped us a lot in the improvement of the manuscript.

We have incorporated all suggestion raised by honorable reviewers in true spirit. Authors feel that the suggestions raised by honorable reviewers have enhanced the readability of the current manuscript. The edited text has been marked by track changes in the manuscript. Following is the point-wise justification of the revision made in the manuscript

Reviewer #2

Comments and Suggestions for Authors

I would like to thank the authors for the very good work done. However, I have some comments.

Comment-1: No mention had been done on the association between flies and summer mastitis on the therapeutic section of the manuscript.

Response: Role of flies in the spread of pathogenic bacteria causing mastitis has been added in the main manuscript. However, many research articles have discussed control measures related to flies in general and not in specefic to mastitis. The manuscript focuses mainly on advanced therapeutic intervention that can be potentially used against mastitis. Therefore, we have restrained ourselves to add basic/general treatment against flies in the main context.

Comment-2: Despite the plethora of published studies, there is a huge lack of information regarding the detection (through HLPC) of specific substances contained in herbs that may have these anti-mastitis effects and, of course, their mechanisms of action. For this reason, I recommend a different section or paragraph describing those important issues. I think it will differentiate your paper and will increase its validity.

Response: A different section in the therapeutic section has been added with respect to identification and role of bioactive compounds in herbal medicine used against mastitis.

Comment-3: Another issue that can complicate diagnostic and therapeutic approaches is vaccination (with commercial product or self-vaccination). It can interfere diagnosis through its producing antibodies (ELISA) and it can "hide" mastitic cases which demand special therapeutic approach. From my personal experience, vaccination made invisible some S. aureus cases and permit this microbe spread in all herd unnoticed before we take action against it. But the damage has done!

Response: A separate section related to immunization against mastitis and its constraints have been added in the main context (Section 7).

Authors are grateful to the honourable reviewers for helping in improving the manuscript.

Reviewer 3 Report

The article reviews the diagnostic, management, and therapy of bovine mastitis. Although it bring a lot of good information to the field, the English is not the most appropriate in some situations. Sentences are to long and need to be short for clear presentation of concepts

I suggest having someone with knowledge of the English language to review the manuscript.

A few comments on wording

Line 24 - replace the word paper by article

Line 32 - replace the word paper by article

Line 38 - replace the word discussed by discusses

Line 51 - replace udder by quarter

Line 51 - replace showed by may present

Line 53 - replace show by may present

Line 56 - replace resulted by results, add reference at the end of the text

Line 62 - what type of mastitis is this refering to? SCM

Line 67 - is accompanied by

Line 69-71 - this is refering to what? mastitis? All types of mastitis?

Line 93 to 97 - this is very confusing. Needs rewording

Line 120 - describe what you mean by conventional method, and advanced diagnostic tools

Line 163 replace diminution with reduction

Line 178 - replace the word papers by articles

In some cases like section 1, the sentences are confusing

Pinzon and cabrera have a really nice research article on the economic impact of mastitis. Should be described in section 4

The English is not very good. Sentences are to long and some times confusing. Especially in section 1

Author Response

Article id-  vetsci-2439808

First of all I would like to thank to all revered reviewers and editor for their critical insight to our manuscript. Reviewer’s suggestions have helped us a lot in the improvement of the manuscript.

We have incorporated all suggestion raised by honorable reviewers in true spirit. Authors feel that the suggestions raised by honorable reviewers have enhanced the readability of the current manuscript. The edited text has been marked by track changes in the manuscript. Following is the point-wise justification of the revision made in the manuscript

Reviewer #3

Comments and Suggestions for Authors

I suggest having someone with knowledge of the English language to review the manuscript.

A few comments on wording

Comment-1: Line 24 - replace the word paper by article.

Response: The word has been replaced as suggested.

Comment-2: Line 32 - replace the word paper by article.

Response: The word has been replaced as suggested.

Comment-3: Line 38 - replace the word discussed by discusses.

Response: The word has been replaced as suggested.

Comment-4: Line 51 - replace udder by quarter.

Response: The word has been replaced as suggested.

Comment-5: Line 51 - replace showed by may present.

Response: The word has been replaced as suggested.

Comment-6: Line 53 - replace show by may present.

Response: The word has been replaced as suggested.

Comment-7: Line 56 - replace resulted by results, add reference at the end of the text

Response: The word has been replaced as suggested.

Comment-8: Line 62 - what type of mastitis is this refering to? SCM

Response: Subclinical word has been added at right place.

Comment-9: Line 67 - is accompanied by

Response: The change has been made as suggested.

Comment-10: Line 69-71 - this is refering to what? mastitis? All types of mastitis?

Response: In the context, sub-clinical mastitis word has been added.

Comment-11: Line 93 to 97 - this is very confusing. Needs rewording

Response: The sentence has been rephrased.

Comment-12: Line 120 - describe what you mean by conventional method, and advanced diagnostic tools

Response: The explanation of conventional and advanced diagnostic techniques have been added in the context

Comment-13: Line 163- replace diminution with reduction.

Response: The changes has been made as suggested

Comment-14: Line 178 - replace the word papers by articles

Response: The changes has been made as suggested.

Authors are grateful to the honourable reviewers for helping in improving the manuscript.

With deep regards

Round 2

Reviewer 2 Report

My comments have been addressed. Thanks!